# The Effects of the Large-Scale Factor on the Integrity Parameters of Monolithic Fire-Resistant Glass



**Marina Gravit [1], Daria Shabunina [1,*], Pavel Stratiy [2], Irina Leonidovna Kotlyarskaya [1] and Maxim Sychov [3]**

[1] Civil Engineering Institute, Peter the Great St.Petersburg Polytechnic University, 195251 Saint-Petersburg, Russia; marina.gravit@mail.ru (M.G.); iravassilek@mail.ru (I.L.K.)
[2] LLC Phototech, 129337 Moscow, Russia; techdir@phototech.ru
[3] Institute of Silicate Chemistry, Russian Academy of Sciences, 199034 Saint-Petersburg, Russia; msychov@lti-gti.ru
[*] Correspondence: shabunina.de@edu.spbstu.ru

**Abstract:** Glass is widely used for the manufacture of the facades and interior glazing of buildings. Glass structures are subject to high fire safety requirements. Two methods are employed in this work: experimental studies of small-sized and large-sized samples and simulations of heating glass structures. The results showed that large-sized samples of monolithic tempered glass, with dimensions of $4250 \times 2000 \times 8$ mm and $2000 \times 3000 \times 8$, that were inserted in a steel frame, if properly installed, provided fire resistance limits of E30/E45 and E60, respectively, for loss of integrity, which proves the influence of the dimensions of the glass panel on the fire resistance of the facade structure. The small-sized samples of monolithic tempered glass with dimensions of $1000 \times 700 \times 8$ mm provided a fire resistance limit of E60 for loss of integrity. A large-sized sample of monolithic tempered glass measuring $4250 \times 2000 \times 8$ mm and inserted into an aluminum frame provided a fire resistance limit of E60, proving the effect of the frame on the fire resistance of the structure. According to the results of several simulations, a conclusion was formed about the possibility of predicting the fire resistance limits of tempered glass based on its thickness and dimensions. During operations, these structures will be able to prevent the spread of fire and combustion products for the required time after the loss of integrity. The results of the study allow for the estimation of the influence of the scale factor on the falling of the glass from the frame in a fire (loss of integrity).

**Keywords:** buildings; glass structure; fire resistance; monolithic fire-resistant glass; loss of integrity; cracking; large-scale factor; heat transfer simulation

## 1. Introduction

Since the beginning of the 2000s, the number of glass structures and elements of buildings has increased significantly. Greater and greater areas in the enclosing wall structures are occupied by glass. Their visual lightness, transparency, accuracy of installation, impact resistance, energy saving potential, UV protection, noise reduction and light transmission make glass structures extremely popular in interior design [1–3].

To ensure a comfortable and safe stay in a building, it is necessary that all building structures are resistant to fires. Glass behaves like a brittle material with a relatively high compressive strength and limited tensile strength; therefore, when exposed to fire for a long time, it breaks into many fragments that can form holes through which fresh air enters and, as a result, the spread of the fire is greatly accelerated [4,5].

The main problem with using glazing systems is the reaction of the glass to temperature fluctuations. Conventional glass used in buildings has limited resistance to fire and breaks within minutes, indicating thermal failure. Special types of glass, called fire-resistant glass, are able to withstand the effects of the thermal and mechanical stresses that occur

during a fire for a certain period of time, preventing the spread of fire and combustion products [6,7].

The mechanism of cracking and subsequent loss of glass significantly affects the dynamics of the fire. In [8], the results of experiments in which a double-glazed window was exposed to a fire are presented. As a result of the study, the rate of heat release, the heat flux distribution, the glass surface temperature, the cracking pattern and the loss of integrity of the glass unit were determined. It has been established that the appearance of cracks is influenced by the thermal load, composition and installation of glass. In [9], glass samples with dimensions of $1200 \times 1200 \times 6$ mm$^3$ and eight different forms of fastening were tested, and the time of initiation and propagation of cracks, the rate of heat release, the temperature of the glass surface and the loss of integrity of the glass unit were studied. Based on the experimental results, the relationship between the form of fastening and the behavior of cracks was discussed. Thermal stress caused by glass temperature differences (from 48 °C to 159 °C) was the reason for the destruction of the curtain facades studied. In [10], thermal stresses on double-glazed windows during a fire were studied; the results can be used to predict where cracks are likely to occur in a glass structure during a building fire. Based on [5,9–11], it has been shown that, depending on the insulating glass unit, the destruction of glass occurs at a temperature between 250 °C and 460 °C.

Float glass is commonly used in the manufacture of insulating glass due to its flawlessly smooth surface, high light transmission and low cost [12,13]. In [14], the characteristics of float glass exposed to radiant heat fluxes are investigated. It was found that at a radiation of 15 kW/m$^2$, there were cracks on the surface of the float glass within the first minute of the experimental study, indicating the beginning of the destruction of the glass panel. In [15], small-scale experimental studies were conducted on tempered fire-resistant glass. The results showed that the critical temperature difference and heat flux of the fire-resistant glass were 340 °C and 46 kW/m$^2$, respectively, which is significantly higher than that of float glass. It is also hypothesized that a frame with higher thermal conductivity can increase the fire resistance of glazing systems. Thus, tempered fire-resistant glass can be used to replace float glass and is widely used in glass facade structures.

There are two types of fire-resistant glass: laminated fire-resistant glass and monolithic (single-layer) glass [16]. Laminated fire-resistant glass is called triplex [17]. The design is based on laminating several glass sheets together with a polymer layer. During a fire, when the temperature of the glass rises to 120–200 °C, the polymer gel layer foams and becomes cloudy, so the glass turns into a rigid opaque structure. When heated, the foaming polymer layer increases in volume by 5–10 times and seals the cracks that form in the glass. In addition, the resulting foam layer insulates the second glass panel from a significant portion of the damaging thermal effects. With further heating, the polymer layers on the second and subsequent sheets of glass begin to foam, protecting the third and subsequent glass sheets from thermal exposure.

In [18], laminated glass was considered for use as a balcony balustrade through an in-depth analysis of empirical data from past fires that occurred in Canada, Australia, and England. It was revealed that the use of laminated glass does not pose a danger from the point of view of the external spread of fire. In [19], fire tests were carried out to check the characteristics of laminated tempered fire-resistant glass; in addition, the coefficient of the glass destruction was checked. It was concluded that in heated laminated glass, the polymer interlayer exhibits complex behavior, and the temperature difference of the laminated glass can be explained by taking into account the thermal effect of the interlayer. In [20], the reaction of laminated glass beams under the influence of fire and a long-term load ($p = 1.15$ kN) is studied experimentally and by modeling. It has been established that laminated glass beams are able to withstand the applied load for 34–51 min before complete destruction in accordance with the limiting deflection rate determined in [21]. In [22], finite element modeling is presented for multilayer and monolithic glasses under thermal action; restrictions that affect the modeling are indicated, namely, various thermophysical properties of materials (e.g., thermal conductivity, heat capacity). According to the results

of these experiments and simulations, it has been shown that the thermal resistance of monolithic glasses is higher than multilayer ones.

The second group of fire-resistant glasses (monolithic single-layer glasses) have a special chemical composition and are tempered under special temperature and chemical conditions. The fire resistance and quality of such glass is entirely dependent on an exclusive technology that is not disclosed. In [23], it is stated that the main problem with monolithic fire-resistant glass is the installation of the frame. Designers and builders often use frame structures with a good appearance and an attractive price, but they do not tend to use steel frames, which are regulated in the glass fire test report. In addition to steel, sometimes an aluminum frame is used for fire-resistant glass, as indicated in [24]; its melting point is about 650 degrees, and the results are usually better, since the aluminum frame is more pliable to the resulting stresses. In [25], a numerical study is carried out to examine the small elements of fire-resistant monolithic glass panels exposed to fire; further validation of the obtained results is demonstrated through modeling with experimentally obtained values. In [5], the effect of glass size on its ability to withstand a fire load is studied; two experiments were carried out with glass sizes of $300 \times 300 \times 6$ mm$^3$ and $600 \times 600 \times 6$ mm$^3$ to test the model using 27 numerical examples with glass panel sizes from $100 \times 100$ mm$^2$ to $1000 \times 1000$ mm$^2$. As a result, the fracture time, stress distribution and crack trajectory were calculated and demonstrated. In [26], experimental studies were carried out on a glazing system with dimensions of $600 \times 600 \times 6$ mm$^3$ under the influence of a fire with a power of 0.16 MW. It was found that the size of the glass panel has a significant effect on the fire resistance of the glass; the fire resistance of a glass panel will decrease as the size of the glass increases or as the aspect ratio of the glass decreases.

In the above studies, glass structures with small dimensions are considered. In practice, continuous glazing is used in modern buildings throughout their entire building height, which is about 3–4 m. To ensure the required fire resistance limit, it is necessary to use tempered and specially machined monolithic glasses and conduct experimental studies to evaluate the fire resistance of glass.

According to ISO 834-1 [27], to normalize the fire resistance limits of glazing systems, the following limit states are distinguished:

- loss of bearing capacity (R) due to the destruction or loss of glass from the test frame, the achievement of the limiting deflection value as determined according to ISO 834-1 [27] or the achievement of the limiting rate of increase in deflection according to ISO 834-1 [27];
- loss of integrity (E) as a result of the glass falling out of the test frame, the appearance of a stable flame on the unheated side of the glass for 10 s or more, the formation of a through hole in the glass and ignition or smoldering;
- loss of thermal insulation capacity (I) due to an increase of more than 140 °C in the average temperature at any point on the surface of the unheated side of the glass or a temperature more than 180 °C above the temperature of the structure before the test;
- restriction of the thermal radiation flux density (W) upon reaching a thermal radiation flux density of 3.5 kW/m$^2$ at a distance of 0.5 m from the unheated side.

According to federal law [28], for window openings in non-load-bearing enclosing structures, the fire resistance limits for loss of integrity are regulated at E15, E30, and E60. Thus, to determine whether the type of glazing complies with the required fire resistance limit for the loss of integrity, it is necessary to conduct experimental studies before the glass falls out of the test frame, a steady flame appears on the unheated side of the glass for 10 s or more, a through hole forms in the glass and ignition or smoldering occur.

According to GOST 33000-2014 [24], the fire resistance limit obtained when testing glass in a standard test frame may not match the fire resistance limit of the same glass installed in a different frame. Therefore, to confirm the possibility of using glass in a particular glazing system, it is necessary to conduct experimental studies using frame structures and methods of fixing glass identified in the technical specification.

FT-1 monolithic glass (Phototech, Russia) is the best option for creating effective fire-resistant translucent structures in facade and exterior glazing. It is subjected to special hardening and machining, which allows it to be used in structures with a fire resistance of up to E60 [24]. For example, FT-1 monolithic glass with package sizes of up to $2120 \times 3450$ mm$^2$ was used in the business center "Chelyabinsk City" in Chelyabinsk, Russia (Figure 1).

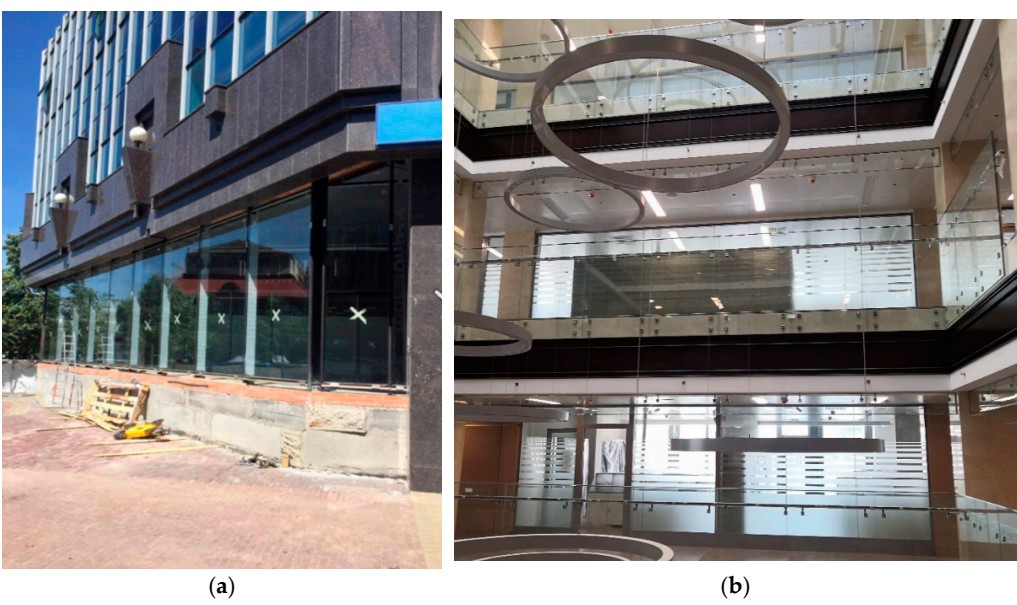

(**a**)                          (**b**)

**Figure 1.** Business center "Chelyabinsk City": (**a**) facade and (**b**) smoke-proof all-glass screen in the atrium. Photo by the authors.

The purpose of this paper is to study the fire resistance limit for the loss of integrity of small-sized and large-sized monolithic, single-layered glass panels using the example of FT-1 brand fire-resistant glass in order to accomplish the following: determine the effect of frame construction on the fire resistance limit of a glass unit, determine the possibility of using steel and aluminum frames with large-sized samples and assess the influence of the dimensions of monolithic single-layer glass panels on their fire resistance limit for the loss of integrity.

The novelty of the study is to establish the dependence of the dimensions of tempered fire-resistant glass and the type of test frame on the fire resistance of the facade structure.

## 2. Materials and Methods

### 2.1. Experiments on Glass Structures

Two methods are employed in this work: experimental studies and simulations of the heating of glass structures. Experimental studies were conducted for two small-sized samples of ФТ-1 tempered monolithic fire-resistant glass without a frame (samples No. 1.1 and No. 1.2), four large-sized samples of ФТ-1 tempered monolithic fire-resistant glass inserted in frames (samples No. 2.1, No. 2.2, and No. 2.3 were tested with steel frames, and sample No. 2.4 was tested with an aluminum frame) and one sample of the FT-1 tempered monolithic fire-resistant glass (sample No. 3.1). The samples considered in the study were tested for loss of integrity (E) according to GOST 33000-2014 [24], defined as the moment when the glass falls out of the test frame, a steady flame appears for 10 s or more on the unheated side of the glass, a through hole forms in the glass and there is ignition or smoldering. Table 1 illustrates the characteristics of the samples.

**Table 1.** Characteristics of the considered samples.

| Sample | Dimensions, mm | Thickness, mm | Frame |
|--------|----------------|---------------|-------|
| Sample No. 1.1 | 1000 × 700 | 8 | - |
| Sample No. 1.2 | 1000 × 700 | 8 | - |
| Sample No. 2.1 | 4250 × 2000 | 8 | steel |
| Sample No. 2.2 | 4250 × 2000 | 8 | steel |
| Sample No. 2.3 | 4250 × 2000 | 8 | steel |
| Sample No. 2.4 | 4250 × 2000 | 8 | aluminum |
| Sample No. 3.1 | 2000 × 3000 | 8 | steel |

According to GOST 33000-2014 [24], the test frame for installing the sample in the furnace ensures that the sample is fixed, while an asbestos gasket is used as thermal insulation between the glass and the test frame, which ensures the consistency of the environment between the glass sample and the frame (Figure 2).

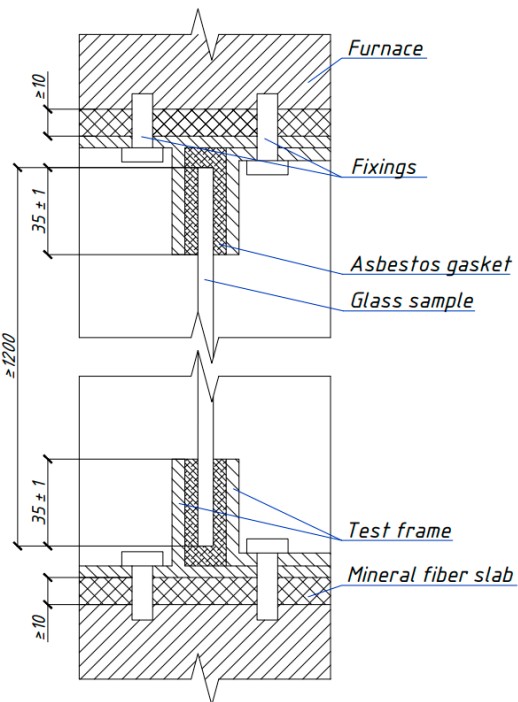

**Figure 2.** Scheme of sample installation in the furnace using the test frame.

The installation diagram of a double-glazed window in a steel and aluminum profile is shown in Figure 3.

Tests were carried out on prototypes to determine the time to reach the limit state in the process of fire exposure under the creation of a standard temperature regime according to ISO 834-1 [27] in the fire chamber of the furnace and characterized by dependence (1):

$$T - T_0 = 345 \cdot \log_{10}(8t + 1), \; where \tag{1}$$

$T$ is the temperature in the furnace corresponding to the time $t$, °C; $T_0$ is temperature in the furnace before the start of thermal exposure $t$, °C; $t$ is the time calculated from the start of testing, min.

The temperature in the furnace fire chamber was measured by TPK-type thermocouples that were installed in such a way that their hot junctions were at a distance of 900 mm from the wall of the fire chamber and 100 mm from the heated surface of the sample.

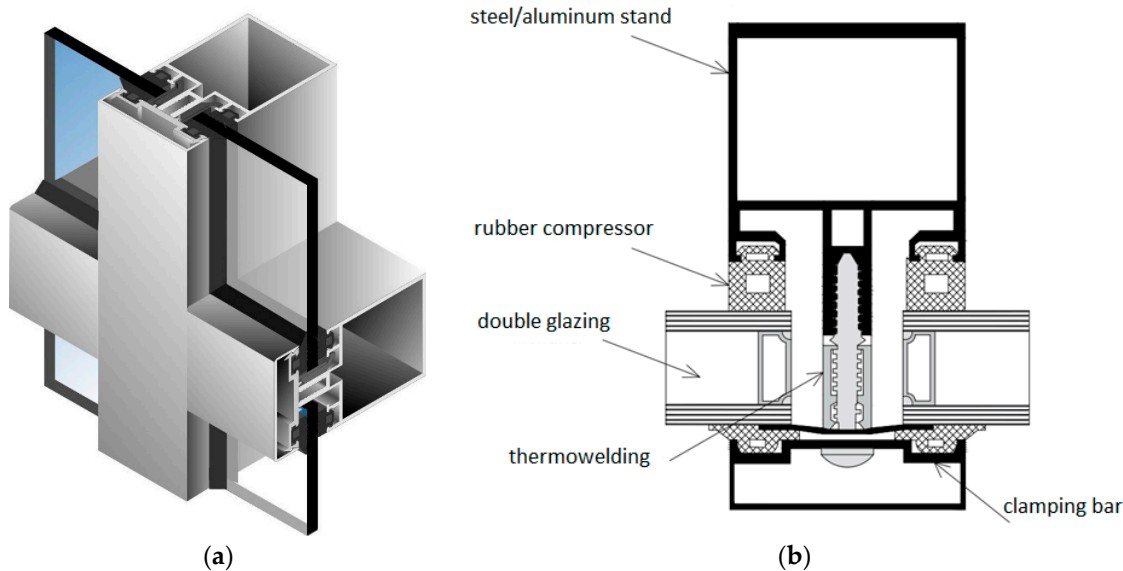

(**a**)  (**b**)

**Figure 3.** (**a**) Visual representation of the insulating glazing in the profile and (**b**) scheme of installation of the insulating glazing in the steel and aluminum profile.

In the small-scale tests, the specimens were tested outside the frame, and, since they were completely inside the fire chamber of the furnace, fastening was carried out using remote composite frames.

A test furnace with a fuel supply and a combustion system was used to conduct the large-scale experimental studies (Figure 4). The equipment consisted of three parts: a combustion chamber, a large-scale calorimeter and a data acquisition and processing system. The combustion chamber consisted of four steel walls, each of which was covered with fire-resistant glass wool that was 15 mm thick. This test method simulated the temperature fields in enclosed spaces under real fire conditions. When conducting large-scale tests of samples of monolithic double-glazed windows, a control was carried out so that the gap between the building opening and the partition frame along the entire perimeter was filled with cement-sand mortar in a ratio of 1:4 and then plastered.

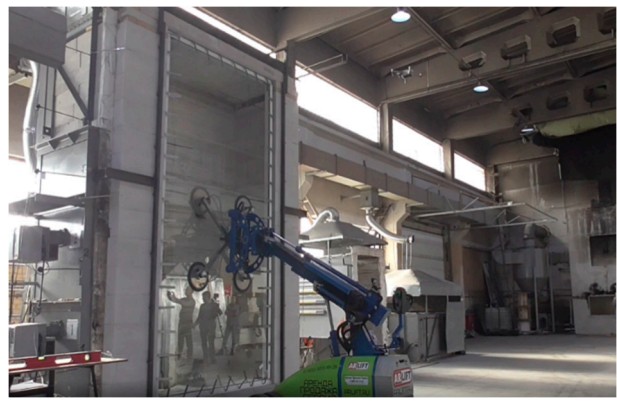 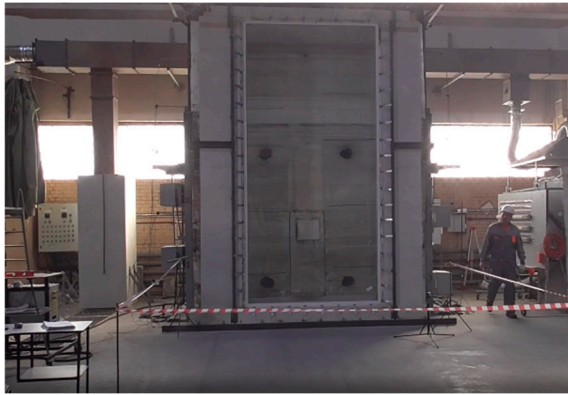

**Figure 4.** Installation of a monolithic single-layer glass panel in the furnace opening. Photo by the authors.

## 2.2. Simulation in SP QuickField

The QuickField software package (SP) was used as a modeling tool (SP Elcut is used as an analogue in Russia). It allows one to specify heat sources in blocks, edges or individual vertices of the model using the finite element method [29]. The SP QuickField (Elcut) has repeatedly been used to solve thermophysical issues. For example, the effect of fire

exposure on modern windows and elements of facade glazing was considered in [30]; it was demonstrated that Elcut allowed for predictions of the behavior of building structures at elevated temperatures and displayed the temperature distributions and stress fields. In [31], a simulation of the heating of offshore stationary platform structures was undertaken, and it showed good correlation with the experimental results; the consumption of mineral slabs for bulkhead construction was predicted, and the parameters of thermal conductivity and the heat capacity of the applied fire protection were specified for temperatures ranging from 0 to 1000 °C. In [32], the results of large-scale fire tests of lightweight thin-walled steel structures for fire protection efficiency are presented using SP Elcut. As a result, temperature–time curves of steel structures in the standard fire regime were obtained, and the simulation showed good correlation with experimental studies.

When modeling the heating problem in QuickField, the heat conduction equation is used, as determined by dependence (2) [33]:

$$\frac{\partial}{\partial x}\left(\lambda_x \frac{\partial T}{\partial x}\right) + \frac{\partial}{\partial y}\left(\lambda_y \frac{\partial T}{\partial y}\right) = -q - c\rho \cdot \frac{\partial T}{\partial t}, \; where \tag{2}$$

$T$ is temperature, °C; $t$ is the time, s; $\lambda$ are the components of the thermal conductivity tensor, W/(m·K); $q$ is a specific power of heat release, W/m$^3$; $c$ is a specific heat capacity, J/(kg·K); and $\rho$ is the density, kg/m$^3$.

Boundary conditions, described by temperature, heat flux, convection and radiation, are set on the external and internal boundaries of the design model. The value $T_0$ is specified as a linear function of the coordinates. The heat flow is described by relations (3) and (4) [29]:

$$F_n = -q_s - \; \text{at the outer borders,} \tag{3}$$

$$F_n^+ - F_n^- = -q_s - \; \text{on internal borders, where} \tag{4}$$

$F_n$ is the normal component of the heat flux density vector; indices "+" and "−" mean "to the left of the boundary" and "to the right of the boundary," respectively, W/m$^2$; $q_s$—is the surface power of the source for the inner boundary; and the outer boundary is the known value of the heat flux through the boundary, W/m$^2$.

Convective heat transfer is determined according to expression (5) [34]:

$$F_n = \alpha \cdot (T - T_0), \; where \tag{5}$$

$\alpha$ is the heat transfer coefficient, W/(K·m$^2$), and $T_0$ is an ambient temperature, K.

The radiation conditions are set on the outer boundary of the model, and the radiative heat transfer is determined by Equation (6) [29]:

$$F_n = k_{SB} \cdot \beta \cdot \left(T^4 - T_0^4\right), \; where \tag{6}$$

$k_{SB}$ is the Stefan–Boltzmann constant, W/(m$^2$·K$^4$); $\beta$ is the absorption coefficient of the surface; and $T_0$ is the temperature of the absorbing medium, K.

To set the design scheme, profile systems that had an air gap (air) and a rubber seal in their design were used. Their thermophysical characteristics (thermal conductivity, heat capacity and density) were variable and depended on temperature (the values were taken from the program reference book). As a frame structure, steel and aluminum profiles were studied, and tempered glass was used as the filling of a double-glazed window. The initial characteristics of steel and aluminum were taken from the program guide, and the tempered glass was obtained from the results of modeling and scientific papers [35–37] (Table 2). It was assumed that the density value would not change during heating. Boundary conditions are presented in Table 3.

**Table 2.** Thermophysical characteristics of materials.

| Material | Density, kg/m³ | λ, W/(m·K) at T, °C | | | C p, J/(kg·K), at T, °C | | |
|---|---|---|---|---|---|---|---|
| | | **20** | **100** | **300** | **20** | **100** | **300** |
| Steel | 7800 | 49 | 49 | 47 | 460 | 475 | 510 |
| Aluminum | 2700 | 225 | 232 | 237 | 900 | 940 | 1030 |
| Glass | 2400 | 0.200 | 0.750 | 1450 | 490 | 560 | 710 |

**Table 3.** Boundary conditions defined in the SP QuickField.

| Name Quantities | Meaning Quantities | Source Information |
|---|---|---|
| Heat transfer coefficient by convection at standard temperature conditions, W/(m²·K) | 25 | [28] |
| Coefficient takeovers surfaces | 0.5 | [27] |
| Initial ambient temperature, °C | 20 | - |
| Time step for calculating the temperature gradient of the structure, s | 60 | - |

## 3. Results

### 3.1. Results of Experiments on Glass Structures

All of the samples were heated from room temperature to failure, according to the standard fire regime curve. As the temperature increased, the glass gradually softened, and micro-cracks formed on its surface. Experimental studies continued until the glass fell out of the test frame, a steady flame appeared on the unheated side of the glass for 10 s or more, a through hole formed in the glass and ignition or smoldering occurred.

#### 3.1.1. Experimental Results of Small-Sized Samples

It was found that the fire resistance limit of small sample No. 1.1, tested under standard temperature conditions, was reached at 72 min due to the glass melting, which corresponds to a fire resistance of E60 (Figure 5). According to the test results for small sample No. 1.2, tested under standard temperature conditions, the fire resistance limit was reached at 65 min due to the melting of the glass, which corresponds to a fire resistance of E60.

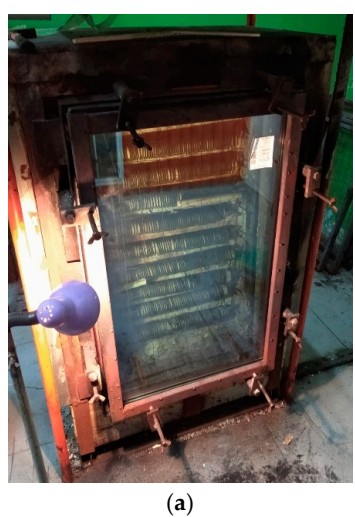
(a)

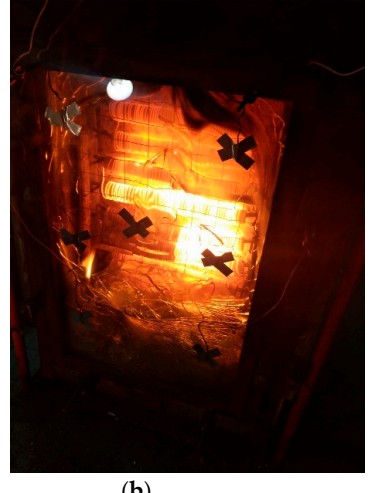
(b)

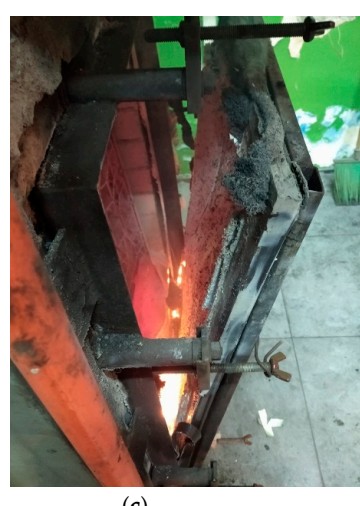
(c)

**Figure 5.** Small-sized sample No. 1.1 (**a**) before, (**b**) in the process and (**c**) after the fire impact. Photo by the authors.

### 3.1.2. Experimental Results of Large-Sized Samples

During the large-scale tests, it was found that the fire resistance limit of sample No. 2.1, tested under standard temperature conditions, was reached at 6 min due to the glass falling out of the test frame, which does not correspond with the required fire resistance limit. When conducting the experimental study of sample No. 2.1, the glass touched the hot steel frame, which led to its destruction. This phenomenon was due to incorrect installation. In the large-scale test of sample No. 2.2, it was found that the fire resistance limit, tested under standard temperature conditions, was reached at 43 min due to the appearance of a steady flame for 10 s on the unheated side of the glass and the glass subsequently falling out of the test steel frame, corresponding to a fire resistance limit of E30 (Figure 6).

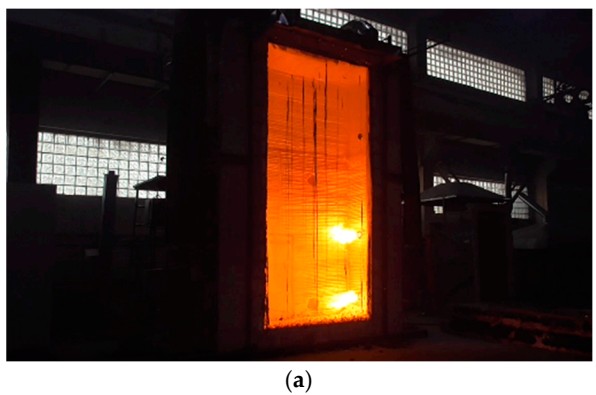
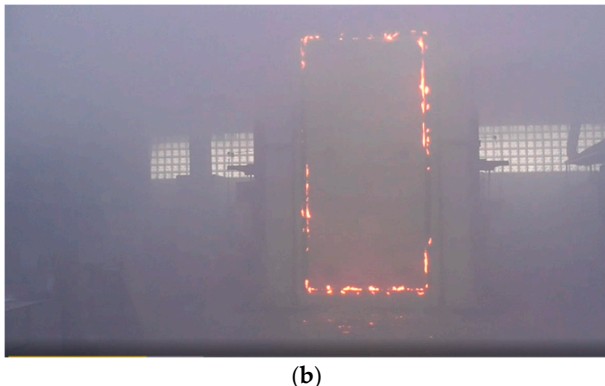

(**a**)                                                                  (**b**)

**Figure 6.** Large-sized sample No. 2.2 (**a**) during fire exposure and (**b**) at the end of fire exposure. Photo by the authors.

In the large-scale test for sample No. 2.3, it was found that the fire resistance limit, tested under standard temperature conditions, was reached at 46 min due to the appearance of a steady flame for 10 s on the unheated side of the glass and the glass subsequently falling out of the test steel frame, which corresponds to a fire resistance limit of E45. It was noted that during the experimental studies of samples No. 2.2 and No. 2.3, the steel frame was deformed first, which consequently caused the glass to fall out of the test frame. In the large-scale test for sample No. 2.4, it was found that the fire resistance limit, tested under standard temperature conditions, was reached at 75 min due to the appearance of a steady flame for 10 s on the unheated side of the glass and the glass subsequently falling out of the aluminum frame, which corresponds to a fire resistance limit of E60.

In the large-scale test for sample No. 3.1, it was found that the fire resistance limit, tested under standard temperature conditions, was reached at 65 min due to the appearance of a steady flame for 10 s on the unheated side of the glass and the glass subsequently falling out of the steel frame, which corresponds to a fire resistance limit of E60 (Figure 7).

The experimental studies showed that with the use of a steel frame, the smaller sample (sample No. 3.1), tested under standard temperature conditions, had a higher fire resistance limit for the loss of integrity of the façade structure than large-sized samples (e.g., sample No. 2.2 and sample No. 2.3) tested under the same experimental conditions. This is because on the surfaces of the double-glazed unit of the large samples, there was a sharp change in the temperature regime from the central part of the glass to the edge, which caused its destruction.

The results of the fire experiment for sample No. 2.4 show that the use of an aluminum frame in the insulating glass structure increases the fire resistance of the structure in comparison to steel frames in terms of loss of integrity.

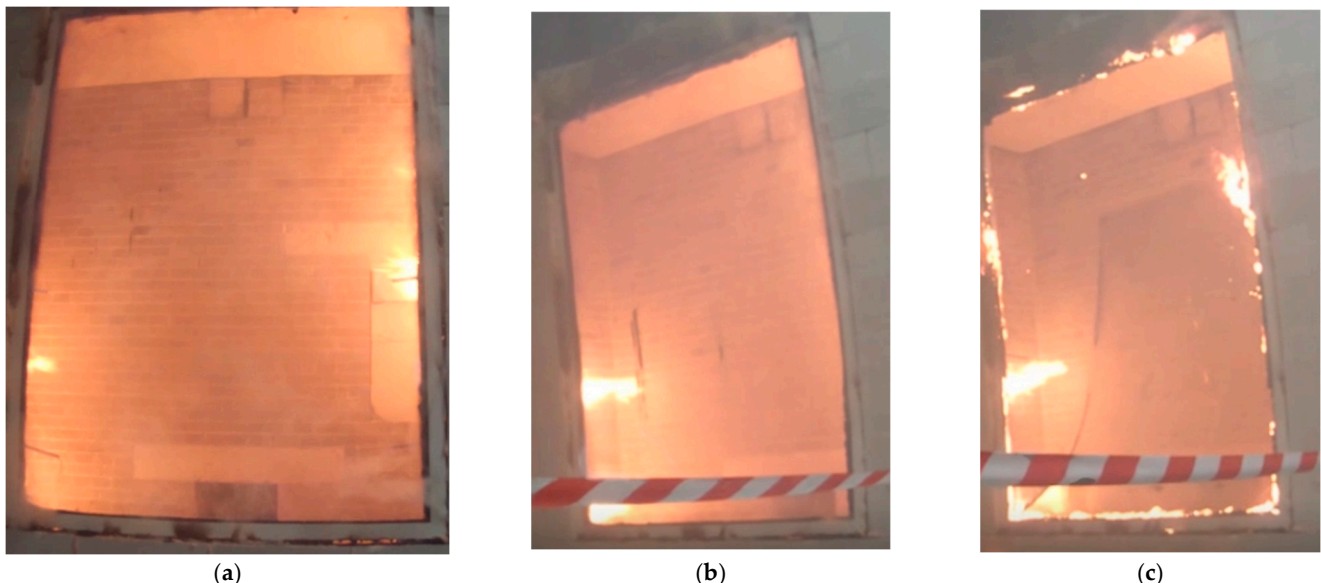

|  |  |  |
|:-:|:-:|:-:|
| (**a**) | (**b**) | (**c**) |

**Figure 7.** Large-sized sample No. 3.1 during the fire impact at (**a**) 48 min, (**b**) 57 min, and (**c**) 60 min. Photo by the authors.

*3.2. Results of Simulations in the SP QuickField*

Using a simulation, visualizations of the heating of the structures of large-sized glazing panels were obtained for panels with dimensions of 4250 × 2000 × 8 mm (samples No. 2.1–2.3) that were inserted into the steel frame at 45 min (Figures 8–10). Due to the large length of the glass (2000 mm), the two parts of the same design are presented from different sides. The temperature value on the unheated side of the glass was 250 °C.

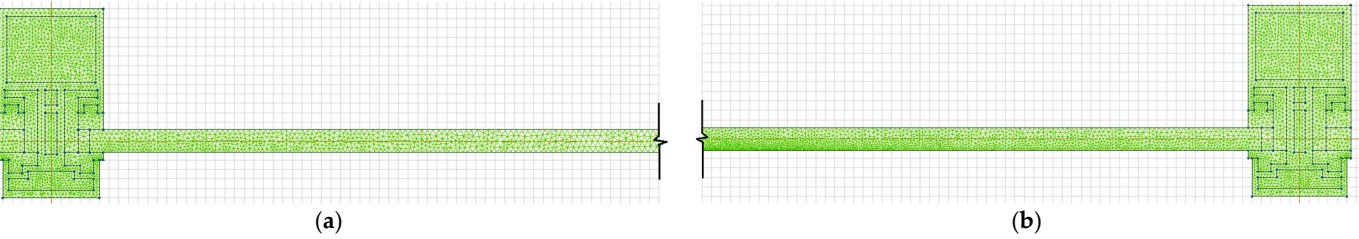

|  |  |
|:-:|:-:|
| (**a**) | (**b**) |

**Figure 8.** Calculation scheme of the (**a**) left side and (**b**) right side of the glazing structure of samples No. 2.1–No. 2.3.

|  |  |
|:-:|:-:|
| (**a**) | (**b**) |

**Figure 9.** Visualization of the heating of the (**a**) left side and (**b**) right side of the glazing structure of samples No. 2.1–No. 2.3.

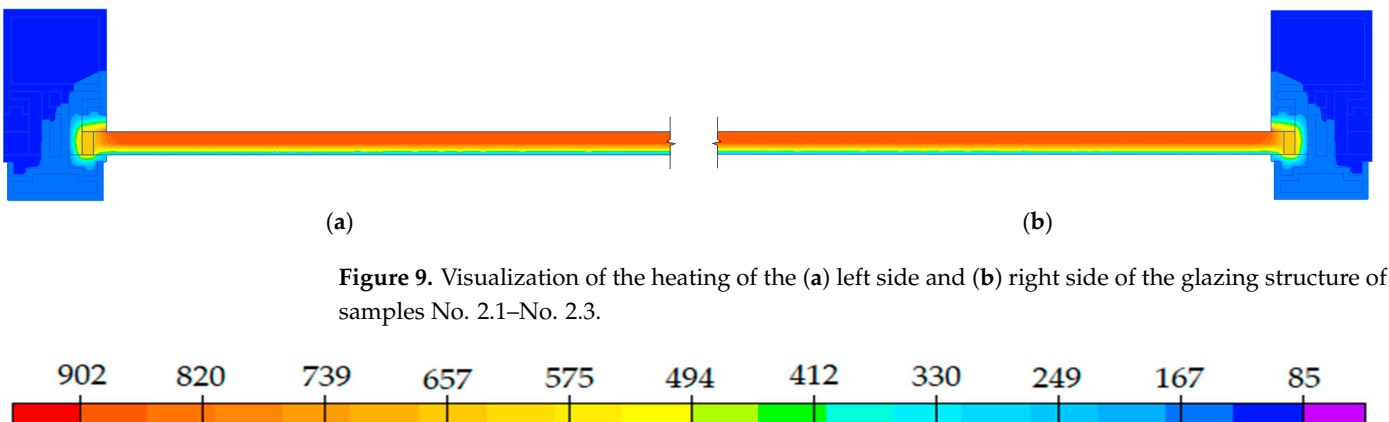

**Figure 10.** Temperature scale for standard fire mode.

The effects of the deformations of the steel profile assembly on the composition of the insulating glass unit of Samples No. 2.1–No. 2.3 as they were put under the action of a thermal load were considered by linking the problems of non-stationary heat transfer and mechanical stresses and deformations (Figure 11). The simulation shows twisting and displacement of 12 mm on the steel frame at the 45th minute from the start of the heating, which would cause the glass unit to fall out.

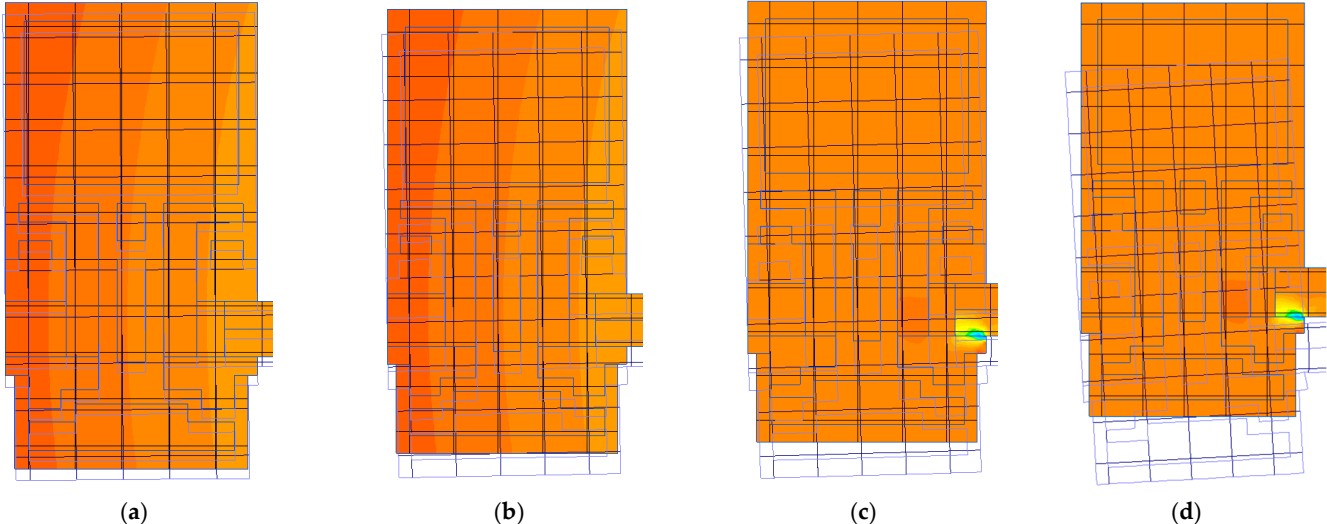

(a)  (b)  (c)  (d)

**Figure 11.** Deformations of the steel profile assembly as part of the insulating glass unit of samples No. 2.1–No. 2.3 while under the action of a thermal load at (**a**) 5 min, (**b**) 15 min, (**c**) 30 min, and (**d**) 45 min.

In the calculation model created in the SP QuickField, the steel frame was replaced with an aluminum one to obtain visualizations of the heating structure of a $4250 \times 2000 \times 8$ mm glazing unit (sample No. 2.4) inserted into an aluminum frame (Figures 12 and 13). Due to the large length of the glass (2000 mm), the two parts of the same design are presented from different sides. The temperature value on the unheated side of the glass was 290 °C.

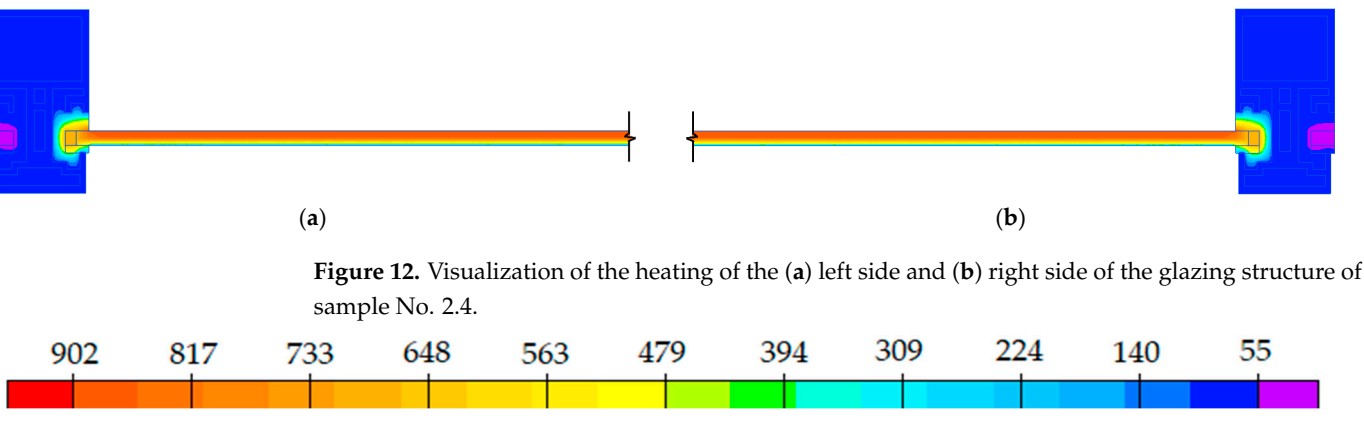

(a)  (b)

**Figure 12.** Visualization of the heating of the (**a**) left side and (**b**) right side of the glazing structure of sample No. 2.4.

| 902 | 817 | 733 | 648 | 563 | 479 | 394 | 309 | 224 | 140 | 55 |

**Figure 13.** Temperature scale for standard fire mode.

The node deformations of an aluminum profile used with the insulating glass unit of sample No. 2.4 as it was put under the action of a thermal load were re-examined by linking the tasks of non-stationary heat transfer, mechanical stresses and deformations (Figure 14). According to the simulation results, it can be seen that on the 45th fire impact, the aluminum frame was displaced by 5 mm; in order to predict the loss of glass from the frame through the simulation, the thermal load was gradually increased, and at 75 min, the aluminum frame reached a displacement of 13 mm, which would have caused the destruction of the double-glazed window.

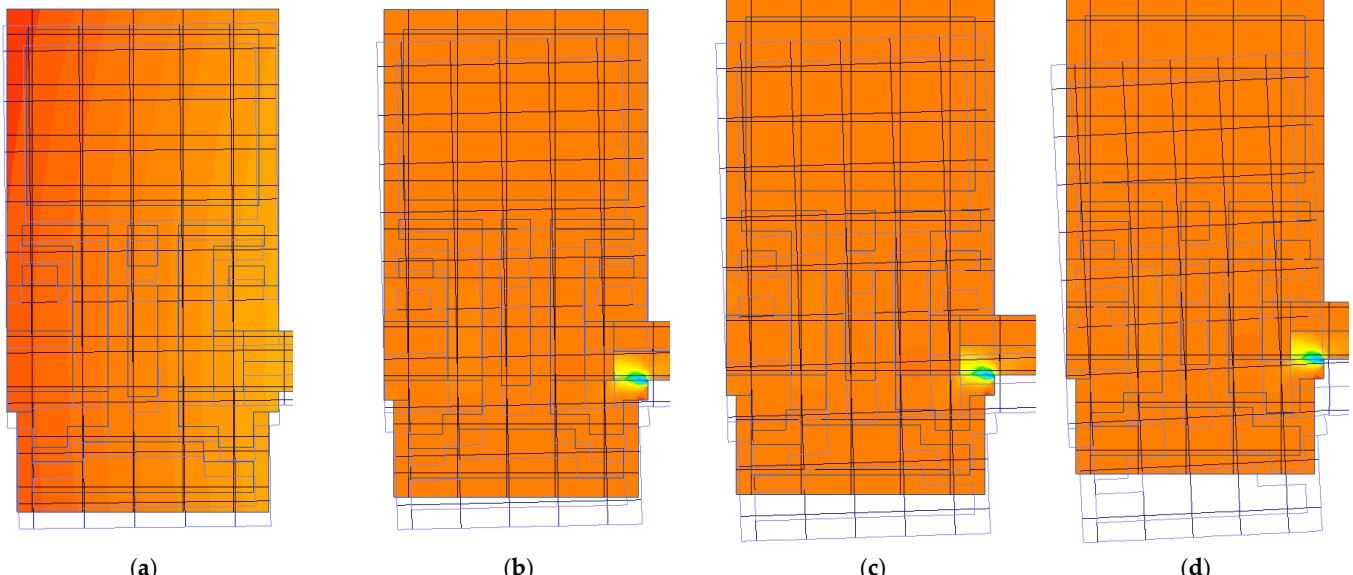

**Figure 14.** Deformations of the aluminum profile assembly used with the insulating glass unit of sample No. 2.4 as it was put under the action of a thermal load, at (**a**) 15 min, (**b**) 45 min, (**c**) 60 min and (**d**) 75 min.

The results of the simulations of the glazing units of samples No. 2.1–No. 2.4 allowed for the determination of the temperature values on the unheated side of the structures when exposed to standard temperature conditions. It was determined that the aluminum frame had a higher temperature value on the unheated side of the insulating glass unit than the steel frames because of the higher thermal conductivity value of aluminum. Glass in a steel frame has higher edge stresses than glass in an aluminum frame, which is observed in these experimental studies and simulations.

## 4. Conclusions

Fire has a serious impact on the integrity of the enclosing structures of buildings and structures, including double-glazed windows.

In this work, experimental studies were conducted on large-sized and small-sized samples of tempered fire-resistant glass as a component of double-glazed windows; these samples were inserted into steel and aluminum frames and placed under the influence of a standard fire mode. It was found that the glass inserted into an aluminum frame had a higher fire resistance limit (despite the fact that the coefficient of temperature deformations in aluminum is two times higher than in steel), because aluminum frames are hollow and thin-walled. The simulation results also confirmed that the steel frames distorted more than twice as much as the aluminum frame when exposed to fire.

According to the results of this experimental study, it was revealed that the fire resistance limit for the loss of integrity of small-sized samples with a steel frame is higher than the fire resistance limit of large-sized samples by 29%. This is due to the fact that on the surfaces of the large, double-glazed samples, there is a sharp change in the temperature regime from the central part of the glass to the edge, which causes their destruction.

It is necessary to test fire-resistant glass for fire resistance in terms of heat-insulating ability and to determine the dependence of fire resistance limits on the chemical composition of glass, manufacturing technology, fastening method and type of installation. This paper shows that, according to the tests, the fire resistance limit for the loss of integrity of small-sized glass panels is E60, and it is E30/E45 for large-sized glass panels. In the event of a fire, FT-1 monolithic glass will be able to maintain its integrity and prevent the spread of fire and combustion products.

The results of the study estimate the influence of the scale factor on the fallout of the glass from a frame in a fire (loss of integrity).

**Author Contributions:** Conceptualization, M.G.; methodology, P.S. and M.S.; software, D.S.; investigation, P.S.; resources, I.V.; data curation, I.V. and M.G.; writing—original draft preparation, D.S.; visualization, M.S. All authors have read and agreed to the published version of the manuscript.

**Funding:** The research was funded by the Russian Science Foundation (RSF) under Grant No. 23-29-00618, https://rscf.ru/project/23-29-00618/.

**Institutional Review Board Statement:** Not applicable.

**Informed Consent Statement:** Not applicable.

**Data Availability Statement:** Not applicable.

**Conflicts of Interest:** The authors declare no conflict of interest.

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
