# Peer review of "The Effects of the Large-Scale Factor on the Integrity Parameters of Monolithic Fire-Resistant Glass"

_fire, doi:10.3390/fire6030114_

Round 1
Reviewer 1 Report
Reviewer's Comments to Author:
This paper studies effect of the large-scale factor on the integrity parameter of 2 monolithic fire-resistant glass by experiment and simulation. It is possible to predict fire resistance limits of the tempered glass. But the novelty and importance of work is not clear and in-depth, so it is suggested to improve in the following aspects:
1 Different constraint condition may lead to different fire resistance limits of glass. In the experiment, how to ensure that the medium between glass and frame can maintain consistency in each experiment?
2.Float glass are used widely in the real application, and what is the different of the effect of the large-scale factor on the integrity parameter of float glass? Some reasons or explanation are necessary.
3. Novelty and importance of work is not properly stated.
4. This article studied the effect of the large-scale factor on the integrity parameter of 2 monolithic fire-resistant glass, but the results focus on the fire resistance limit, other parameters should be considered.
5. The temperature measuring point diagram is suggested to be added.
Reviewer 2 Report
The authors presented the results of research of the integrity parameter of monolithic fire-resistant glass. The manuscript sent for evaluation consists of 15 pages, including 12 pages of text with tables and figures, and 1 pages for references.
Abstract would be to contain all the necessary information, such as methods, results, main statements. Please, add aim of article. Sorry I don't understand the comment in the abstract. The title says "large scale tests" and the abstract says small-scale test". Please clarify.
After reading the text, in general, I assess the scientific quality of the publication as good, but but the quantification of the experimental results is lacking. The visual documentation is illustrative and clear. were temperature profiles taken during the experiments? if so, complete the temperature dependences.
Please specify the novelty of your study in the introduction or conclusion.
- line 151 - the equation is incorrect.
I wish you every success in your future work
Reviewer 3 Report
1. To aid the understanding of the reader, the test criteria mentioned and applied in the study should be described in detail. (It is recommended to fill out a table as much as possible)
2. In Fig.2, you need to input the figures of the drawing.
3. Figure 3, which is a photograph of the exterior of the furnace applied to the experiment, should explain the functional experiment method in which heating is performed.
4. The experiment is an experiment with glass and a frame attached. The experimental equipment in Figure 3 is applicable only to a specific size for which the standard is set. Therefore, in order to secure consistency between experiments and simulations, it is first necessary to secure the validity of simulations by simultaneously performing experiments and simulations on a reference size. This performance is a prerequisite for securing the legitimacy of the research results.
5. Samples applied to experiments and simulations are recommended to be written in tables for better understanding.
6. What do the letters in the photograph of Figure 6 mean? Please use actual experiment data to avoid appearing to be citing external sources that are not real experiment photos. If external sources are used, please substantiate the relevance of the references to this study.
7. This study mentioned two test devices (large and small). In the discussion, the two pieces of equipment should be reviewed separately.
8. As mentioned in the results, the Coefficient of Thermal Expansion values of iron and aluminum are mostly low for Fe, such as AISI_310_SS= 1.512E-05(mm/mm·K) and Aluminum_A356 =2.140E-05(mm/mm·K). Therefore, it could be possible that the Fe frame has a low deformation stress than Al under the condition of thermal expansion for the same experimental temperature change. It needs to be included detailed discussions of whether the difference from the comparison results of iron and aluminum obtained in this study is simply a difference in the hollow structure of the support frame or a difference in the heat transfer rate of the frame. This complementation makes it clear whether this study is a comparison of the heat resistance of glass or a problem of glass stiffness due to frame deformation or heat transfer.
Round 2
Reviewer 1 Report
It can be accepted.
Reviewer 3 Report
It has been confirmed that my questions have been appropriately edited.